# The Neighborhood Contagion Focus as a Spatial Unit for Diagnosis and Epidemiological Action against COVID-19 Contagion in Urban Spaces: A Methodological Proposal for Its Detection and Delimitation

**DOI:** 10.3390/ijerph18063145

**Published:** 2021-03-18

**Authors:** María-Jesús Perles, Juan F. Sortino, Matías F. Mérida

**Affiliations:** 1Department of Geography, Area of Physical Geography, University of Malaga, 29010 Málaga, Spain; 2Physical Geography and Territory Research Group of the University of Málaga, 29010 Málaga, Spain; francis.sortino@uma.es; 3Department of Geography, Regional Geographic Analysis Area, University of Málaga, 29010 Malaga, Spain; mmerida@uma.es; 4Geographical Analysis Research Group of the Department of Geography of the University of Málaga, 29010 Malaga, Spain

**Keywords:** focus of contagion, urban spaces, COVID-19, cartography, spatial pattern, micro-data

## Abstract

The concept of neighborhood contagion focus is defined and justified as a basic spatial unit for epidemiological diagnosis and action, and a specific methodological procedure is provided to detect and map focuses and micro-focuses of contagion without using regular or artificial spatial units. The starting hypothesis is that the contagion in urban spaces manifests unevenly in the form of clusters of cases that are generated and developed by neighborhood contagion. Methodologically, the spatial distribution of those infected in the study area, the city of Málaga (Spain), is firstly analyzed from the disaggregated and anonymous address information. After defining the concept of neighborhood contagion focus and justifying its morphological parameters, a method to detect and map neighborhood contagion focus in urban settings is proposed and applied to the study case. As the main results, the existence of focuses and micro-focuses in the spatial pattern of contagion is verified. Focuses are considered as an ideal spatial analysis unit, and the advantages and potentialities of the use of mapping focus as a useful tool for health and territorial management in different phases of the epidemic are shown.

## 1. Introduction

The absence of references prior to the pandemic raises a lot of uncertainty regarding the spatial model of contagion of COVID-19. The research on the explanatory factors of the contagion and the reasons for its spatial distribution are being simultaneously developed, so the conclusions remain provisional and changing.

As we face the so-called second wave of the Covid-19 epidemic worldwide, it is necessary to reflect on the lessons learned in the fight against contagion during the first wave to identify errors or limitations, and to improve procedures. In the Spanish case, which can be extrapolated to many other countries, the first wave of the epidemic, especially in its early stages, was approached practically blindly regarding the basic tools of knowledge and fight against the contagion. In this situation, the control strategy was based on the severe lockdown of the population, with the obvious economic and social costs for the population.

However, currently, the situation has changed: we have relevant information regarding the importance of aerosols in the contagion, and we have access to a large number of tests for the follow-up of those infected, which provides increasing data on the spatial and temporal behavior of the epidemic. The data show that the spatial distribution of the contagion is uneven at both regional and intra-urban scales [1], although very few researches focus on the accurate detection of the limits of these spatial disparities. More studies are needed to analyze the spatial distribution of the contagion inside the cities, in the neighborhood scale, in which the inter-personal contact occurs in a more intense way; a complex context that lends to multiple inter-relationships between the population and its environment. Chang et al. [2] have recently published a comprehensive study on the virus transmission trajectories in 10 large US cities based on fine-grained mobile phone data. The conclusions emphasize the need to adapt the control measures in order to prevent from a very heterogeneous distribution of the contagion in the urban environment. Faced with this reality, authors such as Rosenkrantz [3] claim the need to incorporate a rigorous spatial approach to the analysis of the pandemic, and an improvement in the quality of the cartographic processes of the epidemiological analysis.

Health authorities habitually use wide areas as spatial units of reference (municipalities, health districts, etc.) to know the evolution of the contagion. However, these spatial units include within them biased and unrepresentative incidence rates, and very different contagion realities, both regarding to their intensity and their modality and origin. As a consequence, the applied measures to fight contagion from these spatial units loses effectiveness. The implementation of generic and homogeneous measures against contagion in urban areas with inner differences in the intensity of contagion along with other aspects, drastically decrease the effectiveness of the measures. They produce a lack of protection of the population in neighborhoods with higher incidence rates, and a fatigue and excessive trust in the inhabitants of little affected areas, who are nevertheless subjected to severe restrictions throughout the period. On a more detailed scale, in many countries, including Spain, the mechanism known as “outbreak tracking” has been enabled to control contagion. The potentials infected related to a specific initial outbreak are identified and put in quarantine in order to stop the chain of contagion from its origin. This monitoring strategy, although very effective, also has drawbacks: it focuses on the traceability of the contagion at the individual level, which makes it a slow and costly process, that is difficult to be maintained during the spread of the contagion or when community transmission occurs.

In this context, the present research proposes, as a complementary strategy to personal tracking, the use of the neighborhood contagion focus as a unit for diagnosing and tracking the contagion in particularly infected local groups. The proposal is based on the trend, tested in this research, that a large part of the outbreaks are reproduced in the same location or in very close sectors in periods of approximately two weeks. This fact is related to the hypothesis which states that those infected that are gathered in a focus at the same moment, produce a transmission network in their family and close neighborhood which will enhance the spread of the disease to new individuals in the same area or adjacent spaces. The selection of the focus of contagion as an epidemiological reference unit allows one to concentrate the resources on the densest and most active focus of contagion, in order to stop the expansion from its roots, that is, at times when the reproduction rates of the focuses are still in a controllable stage.

The contributions of this research are carried out in the framework of the Geo-COVID Cartographic Platform, designed by the University of Málaga to help the decision-makers on epidemiological matters in the city. Geo-COVIDd, funded by the Carlos III Health Institute (Ministry of Sanity, Madrid, Spain), updates and maps epidemiological data in real-time at the maximum level of spatial detail, and supplies decision-makers with them.

### Background on the Use of Cartography and Spatial Analysis for Epidemiological Control

The importance of spatial analysis and detailed cartography for epidemiological diagnosis and control has been confirmed by ancient precedents, such as the pioneering works of the geographer Charles Piquet in Paris, 1832, and John Snow in London, 1854, in relation to the cholera epidemic. More recently, after the development of geographic information systems (GIS), the increasing usefulness of the spatial analysis in the field of health geography has been demonstrated [4,5] for the investigation and management of epidemics [6,7,8,9].

Regarding COVID-19, during 2020, the researches that use GIS to address the analysis of the pandemic have proliferated [10]. A recent publication [11] compiles more than 60 works related to spatial analysis and COVID-19, carried out during 2020, which are organized into five thematic blocks: spatio-temporal analysis, medical geography, environmental variables, data exploitation and web mapping. The GIS are used, among other objectives, to delimit and characterize the areas of greatest danger, to explore possible explanatory factors, and to determine the vulnerability of the population. The most frequent researches are focused on the vulnerability of the population to the epidemic, such as those carried out by Jordan et al. [12], DeCaprio, et al. [13] or Lakhani [14], the latter framed in Melbourne. There are also studies that analyze the socio-economic impact of the pandemic [15]. Sajadi et al. [16] and O’Reilly et al. [17], among others, focus on the explanatory factors that can influence the contagion pattern, such as dynamic (e.g., population flows), and environmental elements (especially climate). Other researches have a methodological approach. For example, some of them have been devoted to data management (big data), its visualization, and its transfer to a GIS [18,19]. Finally, in other cases the research has focused on the detection of new focuses through the analysis of spatio-temporal statistics [20].

The aforementioned studies are presented at different scales, although the regional, national or global are the most common. Adekunle et al. [21] predict the spread of the epidemic for the whole continent of Africa. At a larger scale, GIS have been used to model the spread of coronavirus in several countries and large regional areas, such as East Asia [22], the United States [23], China [24,25], India [26], Pakistan [27] or Iran [28]. At a sub-national level, some works studied the Brazilian states of Sao Paulo, Rio de Janeiro [29], and Bahia [30], whereas at an urban scale, it may be highlighted the studies on social determinants of the expansion of COVID-19 carried out in Buenos Aires and Luján, in Argentina [31], Medellín and Cali, in Colombia [32], or Mexico D.F. [33], complemented with the observations on the epidemic and contagion relationships in the city provided by Hooper [34]. However, the studies that focus on the intra-urban scale are less common. Works can be cited such as those developed by Chadi and Mousannif [35] in Casablanca (Morocco), the study by Gibson and Rush [36] on the suburbs of Cape Town, the research on population density in megacities (London and New York) carried out by Desai [37], or the aforementioned work by Chang et al. [2] about the main cities and metropolitan areas of the USA.

In the light of the cited literature, there is an important lack of detailed studies at the intra-urban scale that may define the spatial pattern of contagion by COVID-19 and their explanatory causes. The main problem that has faced this line of research is the sensitive and confidential nature of the data. Most of the data used by the published cartography use accumulated data by administrative units. This produces a grouping of cases forced by the spatial unit of analysis, which is independent from the natural pattern of contagion. This entails a distortion in the interpretation of the explanatory causes of contagion, and a very important loss of effectiveness in the application of transmission control measures. The use of spatial statistics to identify patterns of elements’ distribution is frequent in geography [38,39,40]. The reflection on the selection of the appropriate scale and the best spatial aggregation method, with the aim of addressing the investigation on the transmission and on the so-called Modifiable Spatial Unit Problem [41], have been aspects previously treated during other epidemics by authors such as Moore and Carpenter [42], or Arsenault et al. [43]. These researches achieve a special meaning in the case of the COVID-19 epidemic.

In the Spanish context, applied researches are being carried out at the intra-urban scale, some of which are presenting preliminary results, in many cases still unpublished. In addition to the research that describes the methodology applied at the present study [44], other ongoing researches can be cited, such as the studies by Miramontes and Balsa focused on Galicia, or the one developed by De Cos, Castillo and Cantarero in Cantabria [45]. All these projects are fed by the addresses of the infected population, that is, the real scale in which the contagion occurs. The study of the vulnerability of the population to the epidemic carried out by Pueyo and Zúñiga in Zaragoza [46], and by Marín and Palomares [47] in Málaga, are also based on the intra-urban scale. On the other hand, Gutiérrez Puebla and García Palomares used geodata from mobile phones to analyze the population movements in Madrid during the most strict stage of mobility constrains. Additionally, other relevant researches are currently in progress in Valencia and Castilla y León. They are being carried out by Olivert Research Group and the GEOTER Research Group of the University of Burgos, respectively.

Therefore, the main objective of the present work is to propose the neighborhood contagion focus by COVID-19 as a basic unit for risk analysis, diagnosis and epidemiological action in urban areas. A specific methodological procedure is also provided to detect, characterize and map contagion focuses. Along with these main objectives, the following partial objectives are addressed in this work.

To identify, at the maximum level of detail and based on disaggregated data, which is the spatial pattern observed in the distribution of those infected by COVID-19 in an urban environment. Specifically, the aim is to examine the level of randomness of the spatial distribution of those infected, as well as to verify the starting hypothesis: the distribution of the contagion by COVID-19 in urban environments presents an uneven spatial distribution, with a higher concentration of infected people in the closest neighborhoods.To typify the morphometric characteristics of the infected agglomerates and to identify the mean distance between cases, in order to use this distance to detect focuses in a predictive way.To propose precise methodological criteria to create focus maps.To apply the methodology to Málaga city (Spain) based on geolocated data of the infected population for seven representative periods of 14 days (bi-weekly), during the expansion and recession phases of the first and of the second waves of the epidemic in the city.To characterize the distribution and dynamics of the contagion in the city in these scenarios.To check out the robustness of the proposed methodology and the concept of contagion focus as a strategic spatial unit for epidemiological diagnosis and action.

## 2. Materials and Methods

### 2.1. Study Case

The research uses the city of Málaga as the study area for the application of the proposed methodology. Málaga is located in the south of Spain, in the Andalusia region, and on the Mediterranean coast (Figure 1). It is the sixth largest Spanish city, with 574,000 inhabitants, reaching one million inhabitants if its metropolitan area is included. It is extended through the coastal plain and the first mountain foothills, forming an urban area of about 63 km^2^.

Economically, it is a very dynamic city, linked to the services sector, with a notable tourist development and it is well connected nationally and internationally. Its structure and internal morphology are not homogeneous. The city brings together a wide variety of building typologies and urban pattern models, which makes it a very good candidate as an area for the contagion pattern analysis. In general terms, it is possible to differentiate the eastern area of the city, with a lower population density and composed mainly of single-family homes in residential zones, with respect to the western and northern areas, which are more extensive and populated, mainly consisting of multi-family buildings (Figure 2).

### 2.2. Sources and Instruments

The proposed approach considers the epidemic as a territorial nature risk [44], according to which the spatial distribution of those infected is the result of random factors (contagion in circumstantial events), together with other structural factors, related to the morphology and functionality of the urban space, which determine the population mobility.

The criteria for the definition of the neighborhood contagion focus concept, which is proposed in this research, are based on the morphometric parameters observed and empirically measured in our study setting. In this work, the anonymized data from the medical records (address and registration dates) of those infected by COVID-19 are used as the basic information source. The daily data of those infected are supplied to the research team by the health authorities of Málaga within the framework of a research project. The authorization for the use of those data for scientific purposes has been previously approved by the Ethics Committee of the Regional Hospital. The data are anonymous, and they comply with the legal framework of the Data Protection Law at the national (Organic Law 3/2018) and the European Regulation (EU) 2016/679 of the European Parliament and of the Council of 27 April 2016. The precision of the data source and its disaggregated character guarantee the rigor of the results, and, given the exceptional of the situation, it should be positively highlighted.

The data are provided by the health authorities on a weekly basis. From the daily data, a grouping has been made into 14-day periods, a time unit widely used during the pandemic at a national and international scale. The objective is to map, for each bi-weekly period, the real scenarios of people simultaneously infected, although in a different phase of the disease. Measurements of the spatial relationships between infected individuals were made in 7 periods of fourteen days (see Figure 3 and Table 1). The selected periods represent different moments of the epidemic evolution in the city (ascending and descending phases of the first and second waves, with peak and valley periods in the trend), in order to sample contrasted contagion scenarios that are representative of the average contagion trend. Table 1 shows several average indicators of the epidemic incidence level in the city throughout the 7 selected bi-weekly periods.

Incidence rates per 100,000 inhabitants in 14 days in the city ranges from 25 in the valley period, between the first and second waves (June), with only 146 infected, a daily average of 28.1 cases and an 0.4 infected/km^2^ (2.3 if only the urbanized area of the municipality is considered), up to the maximum in November, when the highest values of incidence rates were reached (193 infected per 100,000 inhabitants), 1108 cases, 2.8 infected/km^2^ (17.6 if only the urbanized area is considered), and a daily average of infections of 79.1.

In order to estimate the demographic or housing characteristics of the contagion focuses, data from the INE (National Institute of Statistics (Government of Spain, Madrid, Spain) from its Spanish acronym) [48] and the General Directorate of the Land Registry (Electronic Office) [49,50] are used. The greatest level of detailed spatial disaggregation (urban section and cadastral parcel, respectively) is used, which allows a direct correlation with the infected data expressed in the same spatial unit.

The identification and mapping of the focuses of contagion are based on spatial and statistical analysis tools provided by ArcGis Pro software (ESRI, Redlands, CA, USA).

### 2.3. Methodological Procedure

The methodological procedure used in the research is divided into two blocks: in the first one, the general model of distribution of the infected population in the city was analyzed and the starting hypothesis was tested. This hypothesis states that the contagions do not spread homogeneously in the city, but through clusters of cases that generate very different incidence rates by area. This approach leads us to define the concept of a neighborhood contagion focus, and to justify its morphological parameters. On the second block, the methodology for detecting and mapping neighborhood contagion focuses in an urban space was provided and applied to the study case, with the aim of verifying its effectiveness as a basic unit of diagnosis and epidemiological action.

The first methodological stage aimed to discern whether the distribution of those infected made it possible to identify clusters of cases and, thus, to assume the existence of a neighborhood model of contagion. To do so, an average nearest neighbor analysis was applied to the geolocated data, and the R index was calculated. This spatial analysis tool calculates the average distance of all the nearest neighbor distances between cases (average of the distance observed), and it relates this value with the expected mean distance, which corresponds to a hypothetical random distribution of all the cases considered in the analysis in the same area. The ratio between the two distances provides an indicator, the R index, which makes it possible to distinguish between random, dispersed or grouped distribution patterns.

Once the contagion pattern was identified as a distribution model, the selection of the spatial unit of analysis [43] that best collects and represents the natural concentration of the contagion around the focus was addressed.

The potentiality of artificial spatial units (administrative and geometric) to rigorously represent the observed concentration pattern was first analyzed. Firstly, the study area was divided into a regular grid, and the effectiveness of the density map of those infected for the detection of focuses, was tested. To do so, infected people density maps were developed at different grid spacing (number of infected people by regular grids of 1, 2 and 3 has). Moran’s autocorrelation index was applied [51] to check if the squares with highest density are grouped together, and if groups of cases that can be identified as focuses are detected.

Subsequently, the advantages of the neighborhood contagion focus as a basic epidemiological spatial unit were analyzed. The morphological parameters that characterize the clusters were typified in order to use these criteria as a guide to define the concept of focus, and to be able to detect and map new focuses in other spatial and temporal settings. With the aim of establishing these morphometric criteria, the distances between the closest cases (observed distance), the expected distance, and the R index, which were obtained using the average nearest neighbor tool, were applied. Additionally, various basic statistical procedures were calculated, such as frequency and variance analysis. Other indicators used to justify the selection of the neighborhood contagion focus concept and to determine its spatial limits were the daily mobility of the population in its immediate neighborhood. For this purpose, a hypothesis for the assessment of the daily journey from home was formulated, and it is explained in the methods.

Once the neighborhood contagion focus was established as the most suitable basic epidemiological unit, the procedure for detecting and mapping the contagion focuses was systematized. A methodological proposal was designed and it is presented as a result, as well as its application to the scenario, which took place in the city of Málaga (Spain). Finally, once the contagion focuses were detected, a diachronic analysis of their spatial location was developed in order to check their tendency to be reproduced in the nearby areas in a 14 days period, which shows its predictive potential.

## 3. Results and Discussion

### 3.1. The General Distribution Pattern of Contagions in Málaga City

The results of the application of the R index (nearest neighbor average) shown in Table 2 allow us to confirm that the spatial distribution of the contagion in Málaga presents a heterogeneous pattern. Sectors with different patterns are juxtaposed, together with areas that are prone to the concentration of cases. The *p*-values in all the periods analyzed make it possible to discard the null hypothesis of random distribution, just as the R index values lower than 1 indicate the tendency to cluster the data with very high confidence values (Z lower than −12).

It can also be observed how the trend of the concentration index (R) loses intensity as the epidemic spreads through the city and it increases both the number of simultaneous cases in 14 days and the average density of infected people. The Pearson value of the inverse correlation between the number of cases and the value of R is 0.93. It could therefore be concluded that the concentration of cases is more intense in the initial phases, and it tends to homogenize and present a sparse patter in the expansion phase.

In other studies, authors such as De Cos et al. [45], based on the analysis of microdata of infected people equivalent to those used in the present research, observe the same concentration pattern and the trend towards the spatial concentration of cases in Santander and Torrelavega (Spain). Other studies at the urban level, even using coarser data and spatial units, also show the uneven concentration of infected people in urban areas. The explanatory factors that explain this spatial pattern is not clear. The most usual argument relates the incidence of cases to population density. However, this hypothesis is being questioned by authors such as Lall and Wahba [1]. These authors, based on estimates made in New York City, point out that density itself is not a determining factor in the concentration of infections, which are more related to socioeconomic conditions and the quality and access to public services and other facilities. The authors find a direct correlation between the concentration of cases and the dwelling useful area, a point that is also observed by De Cos et al. [45] in Cantabria (Spain). In the analyzed cities, there is a link between the cluster of cases, the average income and the average dwelling size, which are correlated. Chang et al. [2] insist on the great importance of becoming aware of the heterogeneous pattern of contagion in the city when proposing realistic and effective contagion control measures and emphasize the determinant role of areas where the population is concentrated in the transmission.

### 3.2. The Spatial Unit of Analysis. The Concept of Contagion Focus and Its Morphological Parameters

The proposal of the focus of contagion as a preferential spatial unit to express the natural pattern of contagion by COVID-19 in the city is based on the limitations observed in other scales of the spatial representation of contagion. The use of data grouped in administrative spatial units (census section, postal or health district, municipalities, etc.) distorts the natural spatial distribution of contagion and produces a standardization of the statistical results, which can lead to significant errors when diagnosing the epidemic and applying contagion mitigation measures. However, in most of the Spanish autonomous communities, it constitutes the spatial unit of cartographic representation of those infected.

As an alternative to this method, the possibilities of using a regular grid have been checked out to represent the density of the infected and the natural pattern of contagion at a detailed scale. Density maps of infected individuals have been prepared for this purpose by selecting grids of 1, 2 and 3 has. The results of these maps, especially those with the highest precision (1 and 2 has), set out that the small size of the reference unit causes disaggregation of the represented data, and the density per grid is very low. The application of the Moran autocorrelation index to the density data (Table 3) highlights this issue.

As it can be seen in Table 3, it is necessary to use a 3-has grid as a unit of reference to achieve an acceptable spatial autocorrelation and a level of aggregation good enough to represent the natural patter of contagion. However, this cartographic representation unit (number of infected per 3-has grid) has serious limitations in terms of accuracy, and it also introduces the so-called Modifiable Areal Unit Problem (MAUP) described by Openshaw [41]. According to the MAUP, one variable presents unstable results depending on the scale of representation [52]. Despite its limitations, the regular grid is being used as a unit of maximum accuracy to publish official maps of infected individuals (in many cases, misnamed as risk maps) in some Spanish autonomous communities, such as Catalonia.

Due to the limitations of artificial spatial units, a method that allows the contagion focus itself to be used as a spatial unit was designed. The search parameters used to detect contagion focuses was deduced from the morphometric characteristics observed in the cluster of cases. To do this, several density measurements and the R index were applied to the epidemiological scenarios of every period. The resulting parameters are collected in Table 4.

The resulting parameters allow us to observe that as the number of cases in the period increases and the density of the infected people rises, the average distance that separates the closest infected decreases, as does the expected distance. These inverse correlations are confirmed by a Pearson index (0.91 and 0.85, respectively). It can also be observed that the mean separation between cases for all the periods is 151 m. However, this value only partially represents the global trend. On the one hand, the standard deviations of the mean distances observed in each period are high. This fact is related to the presence of a high proportion of 0 values (modal distance) in all the data series. This value corresponds to the cases detected in the same home (contagion of the family, in nursing homes, or in other type of communal houses). The high frequency of modal values leads to a bias of the mean towards low distances. In order to select a more realistic distance between cases, the average distance observed for each period was calculated. Thus, the modal values were deleted (0). A mean distance of 207.2 m was obtained, a value that can be considered as more representative. However, other authors [45] obtained lower mean distances (17.9 m in Santander, and 17 m in Torrelavega). The explanation behind the difference between those results and ours is related to the fact that De Cos et al. [45] used the total number of infected people throughout the period of analysis for the calculation. Thus, they considered people who may not have been simultaneously infected as infected neighbors. In our opinion, for observing the mean separation distance of infected people, it is better to only consider cases of simultaneously infected people (bi-weekly data periods). The accumulation of cases in periods of arbitrary duration draws the contagion pattern scenarios that are dependent on the number of accumulated cases and, therefore, it may be variable and not representative to typify the pattern.

To confirm the representativeness of the distance around 200 m as the mean distance between cases within the focuses, a frequency histogram was prepared with the distances observed between cases in two successive bi-weekly periods in August, during the onset of the epidemic’s second wave in the city (Figure 4).

The histogram provides an accurate orientation of the most frequent distance observed between cases, and of the distance in which there is a change in the morphology of the contagion pattern from a grouped model to a dispersed one. As it can be seen, the most frequent distance between cases is around 100 m, which generate a first level of concentration. At a second level, a change in the concentration is detected at around 200 m between infected people. This value coincides with the mean distance obtained for all the periods analyzed (207 m). The values that are greater than 200 m are much less frequent, and the transition towards a dispersed infected distribution model can be considered. The data obtained for Q1, Q2 and Q3 also confirm this distribution of the distances between the closest affected. As can be seen in Table 4, the average median distance for the different periods is around 100 m (107 m), while Q3 is around 200 m (196 m).

In addition, the use of 200 m as a radius to delimit the most probable scope of transmission of contagion in the neighborhood area, matches with the hypothesis of the population daily mobility proposed in this study. It is based on the idea that people transmit the virus in their daily movements. Leaving aside the movements for working purposes, the population make a series of routes in their neighborhood related to their daily activities (e.g., shopping, walks, access to public services), which act as paths of transmission. In order to know the most likely limits of the contacts within the neighborhood, a standard scenario has been proposed. In this scenario, an individual carries out these activities on 10 min routes. If the estimated speed is 4 km/h, a radius of 600 m can be considered as the walking area. Considering the reduction that the sinuosity of the streets involves in the displacement, a reasonable equivalence between the estimated 600 m and a radial linear distance of 200 m can be proposed. The idea of using the route of those infected in a focus towards crowded places (e.g., commercial or leisure areas) as an orientation route for the spread of the contagions has been used by Chang et al. [2] in their work of the main cities of the USA.

Based on all the proposed arguments, to the idea of establishing a double distance between the cases when detecting and delimiting neighborhood focuses of contagion has finally been considered.

In epidemiological situations comparable to those observed (Table 4) in the study case (contagion density = 8.9 infected/km² of urbanized area, and incidence rate in 14 days = 97.5 infected/100,000 inhabitants), it can be expected that in a focus, the average distance between infected people is around 100 m in the densest areas, and 200 m is the average distance that marks the limit between grouped cases and the areas of the city with a scattered contagion pattern. From these data, it is proposed that 200 m and 100 m are used as the maximum distance between cases to delimit, respectively, focuses and micro-focuses of neighborhood contagion. From the analysis of the results, the following proposal is presented: the neighborhood contagion focus is defined as the area with at least five infected individuals, separated by less than 200 m between them, whereas the micro-focus is the area that includes five infected individuals separated by less than 100 m.

### 3.3. Delimitation of Focuses and Micro-Focuses of the Neighborhood Contagion

Once the concept of neighborhood contagion focus has been defined, the methodological proposal for the detection and cartographic delimitation of neighborhood contagion focuses and micro-focuses consists of the following steps:The application of a cluster analysis with supervised criteria to the geolocated data of the infected people with the aim of generating groupings of at least five cases separated by less than 200 m in focuses, and 100 m in micro-focuses (see Figure 5). The Find Point Clusters tool from ArcGis Pro (ESRI) and the DBSCAN statistical method, which uses a certain distance to separate clusters, have been used.The development of the representative area of the focus for each cluster of cases using the Minimum Bounding Geometry from ArcGIS PRO (ESRI) and smoothing the contour of the polygons obtained by generating a 50 m area of influence around the peripheral cases of each group (Figure 6). To do so, the tool Buffer from ArcGis Pro (ESRI) has been used.

### 3.4. Detection and Mapping of the Neighborhood Contagion Focuses in Málaga

The application of the mapping of neighborhood contagion focuses in Málaga was carried out with representative data of the second wave of the pandemic (August 2020). As it can be seen in Figure 7, the spatial distribution of those infected by COVID-19 during the first half of August is not homogeneous, but it rather confirms the initial hypothesis that proposes the existence of a concentrated and uneven model in the pattern of contagion in the city.

The focuses are located in the northern and western sectors of the city. They are less frequent in the northwest, and they are even hardly manifested in the eastern part, the area with lower population density and with higher socioeconomic level. It is noteworthy that the presence of focuses is considerably lower in the coastal area, a fact that can be even observed in the western part, despite the high number of focuses in that area. In relation to this, some authors such as Coccia [53] or Zoran et al. [54] have revealed the worst transmission of the virus in coastal environments, which are usually less polluted and more affected by coastal winds. On the other hand, as pointed out by De Cos et al. [45], in Santander and in Torrelavega, no focuses of contagion are observed in the historic–commercial center of the city, where the residential uses are minor.

Table 5 shows the parameters of the epidemic in the sectors where the focus have been delimited compared to areas in which the contagion pattern is dispersed. As it can be seen, the values in both areas are significantly different.

The average density of infected people for the municipality is not representative of the reality observed, neither in the area classified as a focus of contagion, nor in the rest of the study area. The density of infected people in the focuses is 32 times greater, even if only the urbanized area of the municipality is considered (63.04 km^2^). The incidence rates per 100,000 inhabitants are also very unequal depending on if they are exclusively estimated for the population within the focuses. In that case, the rate is around 10 times higher. The contagion rate of the virus within the focuses is also significantly higher (4.31) than that observed in the sectors with dispersed transmission (1.21).

Once the spatial differences in the intensity of the contagion were clearly reported, it was examined whether those infected within a focus detected during the first period generate contagions in the same area during the second period. To do so, a map of focuses has been developed in two successive bi-weekly periods in August.

Figure 8 shows the focuses identified during the first and the second periods in August. Apart from the creation of new focuses in new locations during the second period, the focuses of the first period tend also to remain during the second period. Different models of the spread of contagion can be observed: some focuses from the first period remain active in the second period at the same location and their dimensions are similar. Other focuses not only remain in the same area, but instead they are also bigger during the second period. A third type of focus remain in a very close location, showing a slight displacement towards the neighboring areas. These facts seem to prove that a large part of the contagion in a focus is enhanced by the direct contact of the population with the neighbors infected during a previous period, since they coincide in common areas in the neighborhood. Those infected during the first period in a focus are responsible for the transmission of the virus during the second one. This fact gives real-time focus maps, which have a very powerful predictive dimension.

The quantified results of this diachronic spatial analysis of the evolution of contagion are shown in Table 6.

As can be seen in Table 6, there are 20 intersection areas among the focuses of the first and the second periods of August, which generate a common space of 135.5 has. Altogether, it is observed that 71% of the surface considered as a focus in the first period remain as a focus during the following phase; therefore, once 14 days have passed from the first contagion in the area, a focus is reproduced with new infected people in the same space. In another 22 points of the study area (494.3 hectares) it is also found that, in the second period, focuses have been generated less than 250 m away from the original focuses, which allow us to deduce that a transfer of the contagion to adjacent spaces is taking place. Only 27% of the area considered as focuses during the second period are not related to any focus of the first period. These results clearly confirm the robustness of the neighborhood focus as a spatial unit of transmission, and it shows its predictive capacity. Based on the analysis of concentric distances between the cases of the first and the second waves, De Cos et al. [45] reach a similar conclusion about the spatio-temporal permanence of the contagion. Additionally, our results show that the 21.3% of the new positives during a 14-day period in the study case are located within a focus previously detected. This proportion reaches 58.2% if we include a 250 m radius area around those focuses. Therefore, the proportion of new positives is significantly lower in new focuses (30.8%) than in the surroundings of focuses already detected.

On the other hand, the expansion of the epidemic in the city during the second period of August (Table 5) generated, in 14 days, an increase in the incidence in the municipality (from 68 to 182 infected per 100,000 inhabitants), and a similar increase in the number of focuses (from 22 to 51). Consequently, this was accompanied by an increase in the area occupied by focuses (Table 6). In this situation of density increase, the spatial unit of the focus begins to lose spatial accuracy, which can be compensated by their combination with the micro-focuses (distances between cases less than 100 m).

In Figure 9, it can be seen how the micro-focuses delimit those spaces of greater incidence within the focuses, and their delimitation helps to classify the contagion with greater spatial accuracy.

However, when analyzing the duration of the micro-focuses in two successive periods, it can be observed that this micro-unit is more ephemeral. From the eight micro-focuses detected in the first period, only three still remain during the second one (Figure 10). These results seem to indicate that the transmission does not always occur in the closest area of the neighborhood (closer than 100 m), but it rather expands to a buffer of 200 m, a zone that coincides with the daily walking average area proposed as the focus boundary.

However, the results obtained by De Cos et al. [45] show how the mean distance between the cases of the first and second waves is less than 70 m in 75% of the cases. This distance between cases is smaller than ours, but it does not contradict our conclusions, since it is explained by the fact that De Cos et al. [45] used the total amount of cases from the first and the second phase, so the pattern is denser and the expected average distance between neighbors is shortened. The diachronic analysis carried out in our work corroborates the validity of the morphological parameters estimated to size the contagion pattern and the concept of focus and micro-focus.

In addition, the use of the time sequences offers great possibilities to significantly improve the effectiveness of contagion control strategies. If the detection of a focus is carried out in real-time, the resulting cartography has a powerful value as a predictive tool to stop the transmission. Once the focus has been detected, there is a period of 10 to 15 days to carry out a screening campaign on the population within the focus, aiming at detecting asymptomatic positives in the early stages of the disease. Therefore, the chain of contagion can be stopped from its root, both in space and time. This mechanism for tracking groups with higher incidence, compared to other areas of the city, is a key tool to stop contagion more effectively. Chang et al. [2] and De Cos et al. [45] propose the free provision of epidemiological tests to the population of neighborhoods with a high risk of contagion as a way to stop transmission.

The possibility of mapping focuses in real-time is also a very powerful tool to guide mobility and to inform the population of several contagion areas of the city. The delimitation of areas with different contagion risk allows the creation of a cartography, such as the map shown in Figure 11. That map displays the focuses and micro-focuses, and they are used to classify the intensity of the contagion in the city by zones. This allows one to know the number of people infected in each focus at that time, and so, to estimate the risk in their surrounding area and to adjust the level of alert, as well as to implement measures addressed to mitigate the severity of the situation, especially in crowded places (e.g., shops and public services).

We agree with Chang et al. [2], who detailed that a spatial analysis allows for the application of accurate measures in the short-term, which can substantially influence infection rates (e.g., reducing the capacity in shopping centers at a certain moment, or the appropriate management of the population’s mobility in the most affected areas). Maps such as the one included in Figure 11 uses the semiotics of a traffic light to show the number of contagion in each area, and its respective level of alert.

## 4. Conclusions

The obtained results allow an advance in the knowledge of the epidemic’s spatial pattern of transmission in urban areas, and they open a way to improve the understanding of its explanatory causes. In addition, they highlight the advantages of using the neighborhood contagion focus as a diagnostic unit against the epidemiological risk.

In particular, the proposed methodology and the results obtained allow us to conclude:The spatial distribution pattern of those infected shows that the transmission of COVID-19 tends to be grouped, and that the incidence of contagion in urban areas is uneven. All the parameters analyzed show higher rates within focuses. Specifically, the contagion rates are 3.5 times higher, the accumulated incidence is 10 times higher, and the infected density 25 times higher. Consequently, the calculation of the mean values of the aforementioned parameters in urban areas, which are usually published in the official statistics, are poorly representative of the real situation in the city. The cluster of cases have enough robustness to assume that the origin of the contagion, both in simultaneous and successive periods, comes from the closest neighborhood area.The maps that display the density of the infected people, or the number of cases in the artificial spatial units, administrative units or regular squares, have serious limitations in detecting the real concentrations of the infected people at the intra-urban level. They tend to alter the representation of the natural pattern of contagion, which misleads both the epidemiological diagnosis and the measures against contagion. Nevertheless, the use of disaggregated data of the infected people constitutes a remarkable and exceptional contribution to the analysis of the contagion pattern.The analysis of the spatial pattern of those infected and their spatio-temporal evolution at a detailed scale has allowed for the deduction and typification of the morphological and temporal characteristics of the cluster of cases. Based on these data, the concept of focus of contagion neighborhood and the methodology for its detection has been proposed. It is important to mention that the pattern analysis has been performed with accumulated data of 14-day periods. This makes it possible to observe synchronous cases of infected people, instead of accumulated cases which are not necessarily connected.The focuses include infected people separated by the average distance between the closest cases if that distance is shorter than 200 m. Beyond this distance, the pattern of the infected people is dispersed and the odds of the neighborhood contagion focus that remains in the area decrease. In Málaga, the contagion rate decreases from 4.31 cases within the focus, to 1.21 in the area of dispersed contagion. At a second level of analysis, the infected are closer within the focuses, especially up to 100 m between cases.The mapping of focuses in real-time and in bi-weekly periods provides a snapshot of the situation based on the number of cases (symptomatic and/or asymptomatic). This enables one to apply a screening strategy in the neighborhood at that time, and thus, to apply the massive immobilization of asymptomatics. That strategy for monitoring the neighborhood offers a complement to individual tracking during an outbreak. The tracking of a group allows to one detect and immobilize more cases of asymptomatics in less time and with less effort. Given the non-linear trend of the reproduction rate, this control mechanism is especially efficient if it is set in the early stages of the epidemic expansion in an area.The neighborhood contagion focus is the basic spatial unit that represents the natural distribution of the contagion in intra-urban areas; it delimits spaces with differentiated epidemiological diagnosis and different needs. It meets the criteria proposed by Arsenault et al. [43] to select a geographic unit when using epidemiological data: intra-unit homogeneity, variable size of the population of the area, variable extension of the area, and the option of obtaining relevant results. Therefore, its potential to stop contagion, especially in the early or late stages of the epidemic’s expansion, make it a strategic tool to control contagion.This study, which characterizes the spatial pattern of contagion of the SARS-CoV-2 virus, does not only describe a useful control instrument for facing the waves until the immunity of the population can be found thanks to vaccines, but it also offers a methodology that can be used for the prevention and control of any similar epidemic or pandemic.The methodology designed in this work to detect and delimit the location of the focuses opens a new line of research in which the creation, expansion and evolution of the focuses of SARS-CoV-2 and the spatial pattern within the area on which they occur are related. The observation and analysis of the spatial patterns’ connections with the spread of focuses can help to understand some underlying causes of contagion not yet studied at a detailed scale.Since the COVID-19 crisis is a biological risk, it is necessary to apply a spatial approach, in which the mapping of focuses is a key prevention tool to prioritize and intensify measures in the most dangerous neighborhoods, as well to guide information and vaccination strategies. This approach, entitled “geoprevention” by De Cos et al. [45], is essential in order to face the foreseeable successive waves with more guarantees for the economy and the population’s way of life.

## Figures and Tables

**Figure 1 ijerph-18-03145-f001:**
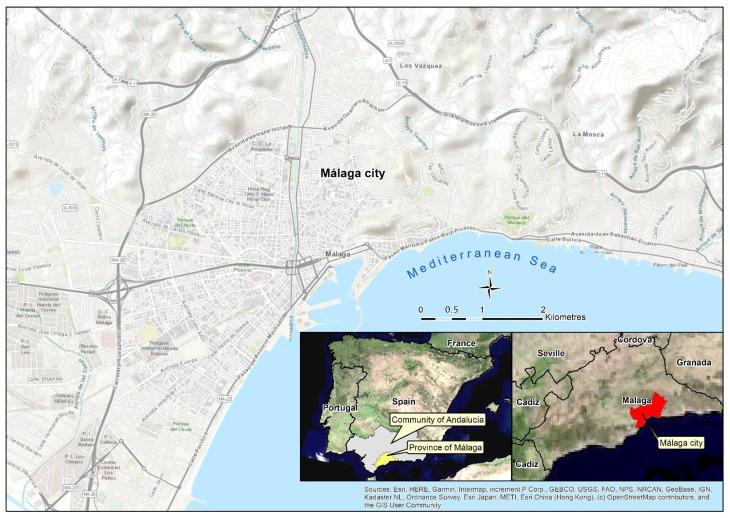
Location and general features of the study area (Málaga, Spain).

**Figure 2 ijerph-18-03145-f002:**
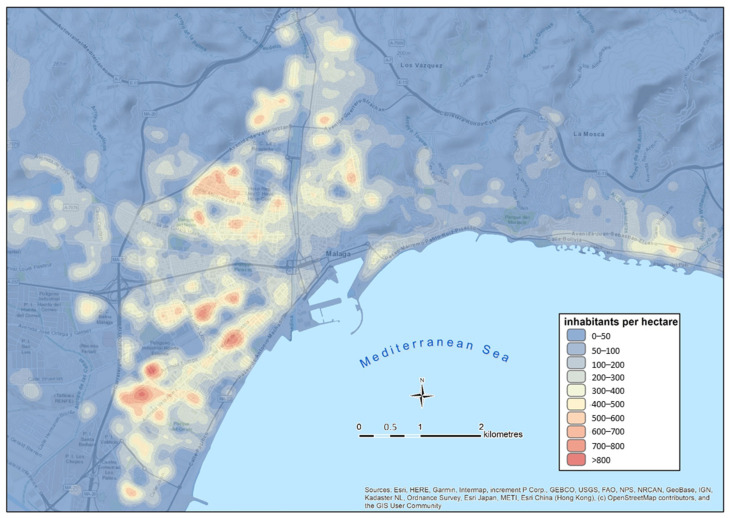
Population density map of the city of Málaga.

**Figure 3 ijerph-18-03145-f003:**
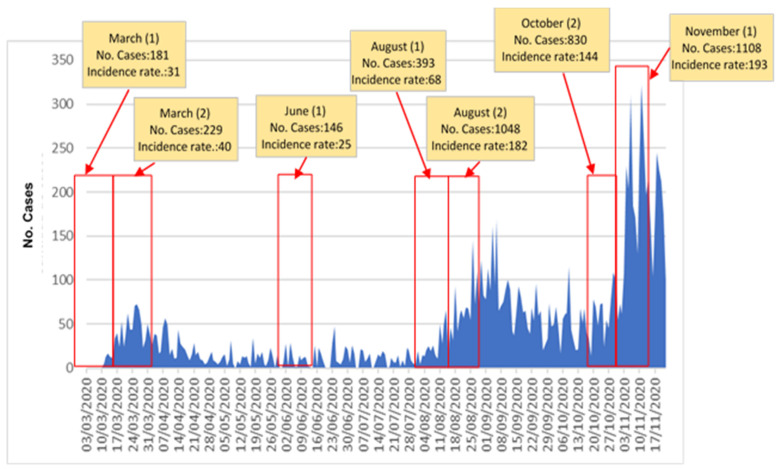
Weekly evolution of the number of people infected in the city (March to November 2020) and selected periods for the study of the spatial pattern of contagion.

**Figure 4 ijerph-18-03145-f004:**
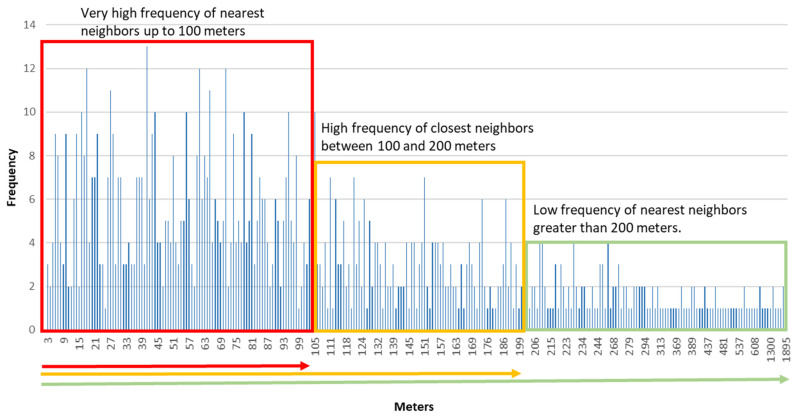
Frequency histogram of the mean distances observed between cases (nearest neighbor). August 2020.

**Figure 5 ijerph-18-03145-f005:**
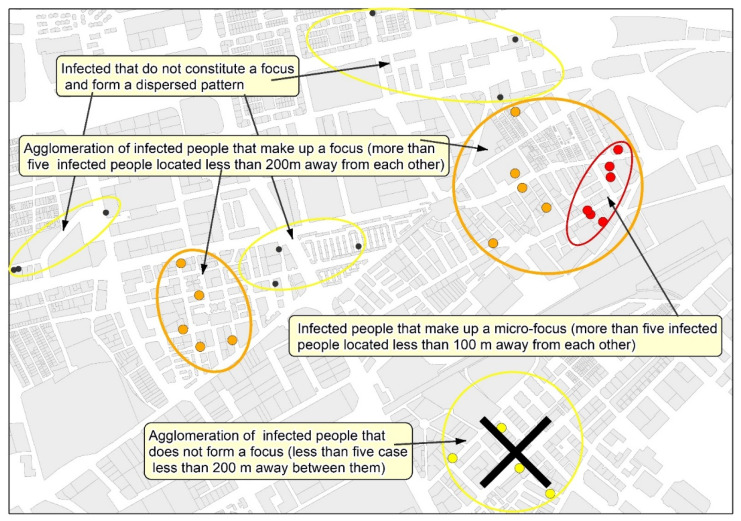
Application of a cluster analysis for the detection of cases according to supervised grouping criteria.

**Figure 6 ijerph-18-03145-f006:**
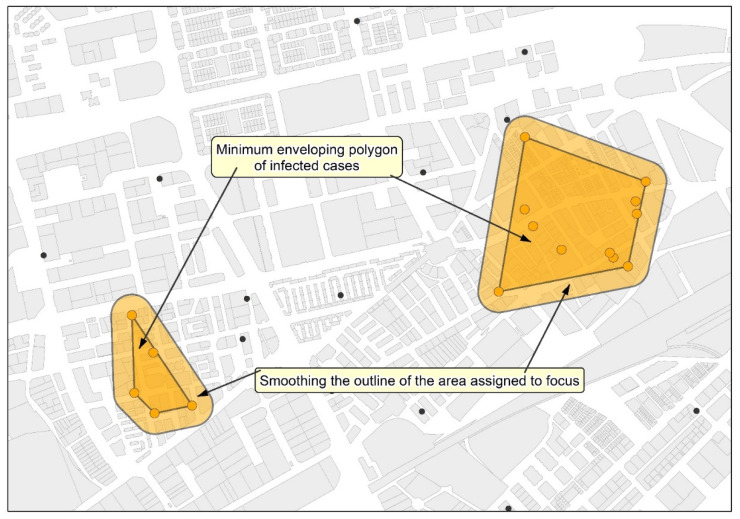
Assignment of the area corresponding to the different infected people (focus) groups and smoothing the limit of the focus.

**Figure 7 ijerph-18-03145-f007:**
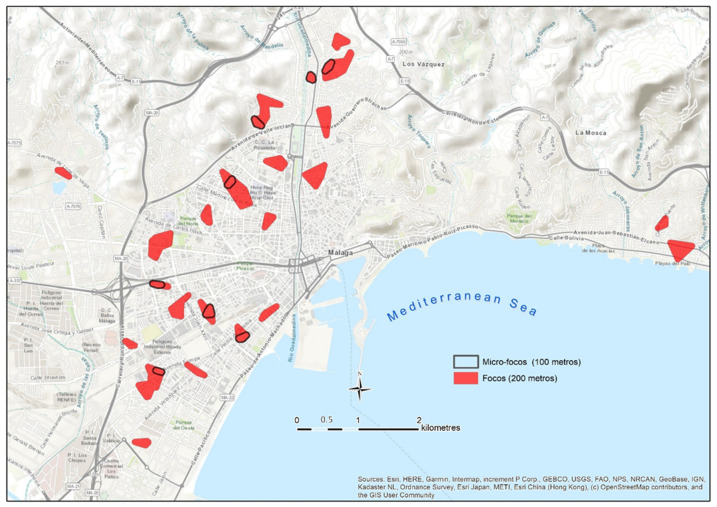
Distribution map of the contagion focuses in the first half of August.

**Figure 8 ijerph-18-03145-f008:**
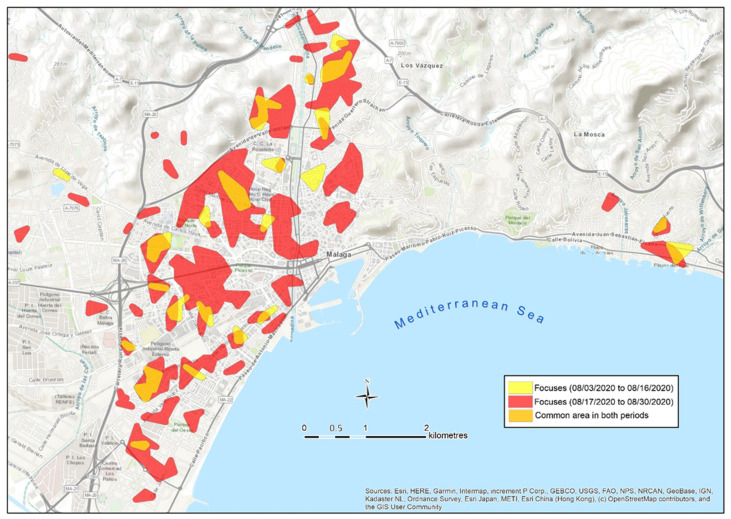
Superposition of the contagion focuses in Málaga in two successive bi-weekly periods during August 2020.

**Figure 9 ijerph-18-03145-f009:**
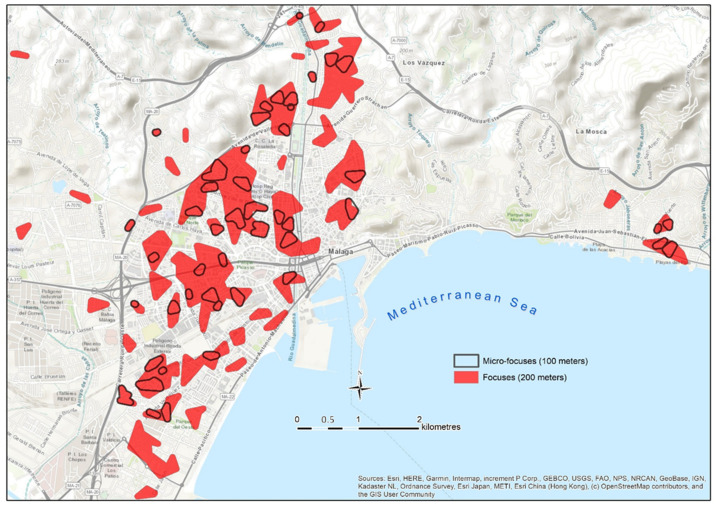
Superposition of micro-focuses and focuses of contagion in Málaga for the second period of August (17 August 2020 to 30 August 2020).

**Figure 10 ijerph-18-03145-f010:**
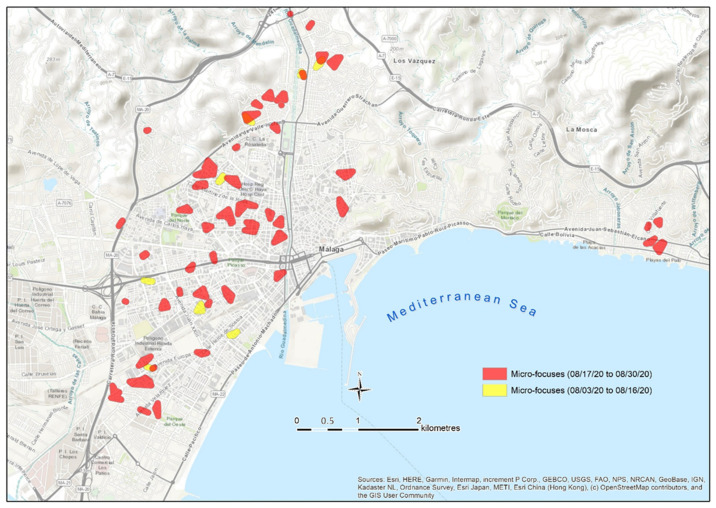
Superposition of the contagion micro-focuses in the city of Málaga in two successive bi-weekly periods during the month of August.

**Figure 11 ijerph-18-03145-f011:**
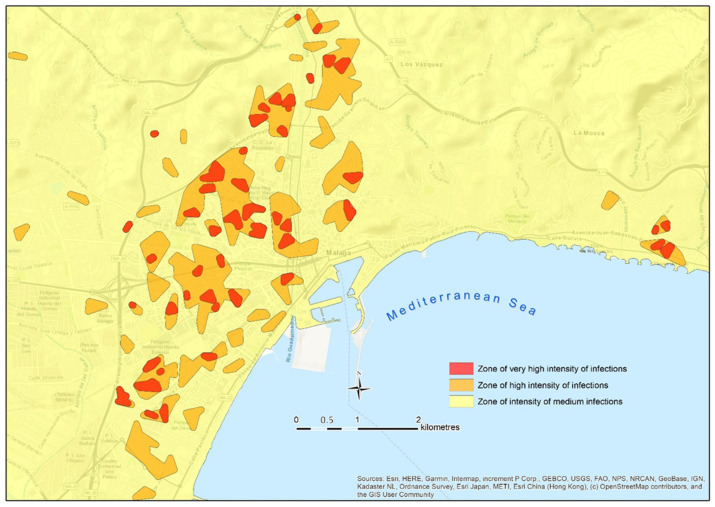
Gradation of the intensity of the contagion from the delimitation of the focuses and micro-focuses.

**Table 1 ijerph-18-03145-t001:** Average indicators of the epidemic level of incidence in the city for the different selected analyzed periods.

14-Day Periods	Average Number of Daily Cases	Average Density of the Infected People Per km^2^ (Municipality)	Average Density of Infected People Per km² (Urbanized Area)	Incidence Rate in 14 Days/100,000 Inhabitants	Total Number of Cases for the Period (14 Days)
March (1)	12.1	0.5	2.9	31	181
March (2)	14.3	0.6	3.6	40	229
June (1)	10.4	0.4	2.3	25	146
August (1)	28.1	1	6.2	68	393
August (2)	74.9	2.7	16.6	182	1048
October (2)	59.3	2.1	13.2	144	830
November (1)	79.1	2.8	17.6	193	1108

**Table 2 ijerph-18-03145-t002:** Concentration values of the infected according to the closest average method in the different periods analyzed, and the general parameters of the epidemics’ incidence in each period.

R Index Value (Average of the Nearest Neighbor)	Z Value (Average of the Nearest Neighbor)	*p* Value (Average of the Nearest Neighbor)	Average Density of the Infected People per Hectare (Municipality)	Average Density of the Infected People per Hectare (Urbanized Area)	Total Number of Cases for the Period (14 Days)
0.5	−12.2	*p* < 0.0001	0.005	0.029	181
0.5	−13	*p* < 0.0001	0.006	0.036	229
0.46	−12.2	*p* < 0.0001	0.004	0.023	146
0.37	−23	*p* < 0.0001	0.01	0.062	393
0.2	−44	*p* < 0.0001	0.027	0.166	1048
0.32	−37	*p* < 0.0001	0.021	0.132	830
0.2	−45	*p* < 0.0001	0.028	0.176	1108

**Table 3 ijerph-18-03145-t003:** Results of the Moran autocorrelation index for different spatial units of density (number of infected per ha).

Spatial Unit Size (has.)	Spatial Unit Size (m^2^)	Moran Index	Z Score	*p* Value
1 ha *	10,000	0.006	0.9	0.3
2 ha	40,000	0.015	1.5	0.1
3 ha	90,000	0.12	8.1	0.0

* One hectare (ha) equals 10,000 m^2^ (100 × 100 m).

**Table 4 ijerph-18-03145-t004:** Summary of the basic statistical data for the different epidemiological scenarios.

14-Day Periods	Incidence Rate in 14 Days/100,000 Inhabitants	Total Number of Cases for the Period (14 Days)	Average Density of Infected People Per km^2^ (Municipality)	Average Density of Infected People Per km^2^ (Urbanized Area)	Average Distance Observed (Meters)	Standard Deviation of Mean Observed Distance (Meters)	Average Distance Observed without “0” Values (Meters)	Quartiles (Meters)			Modal Distance Observed (Meters)	Average Expected Distance (Meters)
								Q1	Median (Q2)	Q3		
March (1)	31	181	0.5	2.9	232.2	413.5	304.6	41.0	148.4	276.5	0	442.5
March (2)	40	229	0.6	3.6	229.9	374.6	253.1	84.4	146.4	276.8	0	443
June (1)	25	146	0.4	2.3	237.7	540.8	369.2	0.0	174.6	320.1	0	506.2
August (1)	68	393	1	6.2	139	217.6	178.4	13.8	77.6	173.4	0	375
August (2)	182	1048	2.7	16.6	67.8	128.5	107.9	0.0	49.3	105.8	0	241.2
October (2)	144	830	2.1	13.2	78.04	133.1	121.8	0.0	62.7	113.6	0	243.8
November (1)	193	1108	2.8	17.6	72.2	291.3	115.4	0.0	52.9	111.4	0	2.47
Average value for all periods	97.5	562	1.4	8.9	151.0	299.9	207.2	19.9	101.7	196.8	0	322.0

**Table 5 ijerph-18-03145-t005:** Data of the incidence of the epidemic inside/outside the contagion focuses.

**Periods**	**Municipality of Málaga (Urbanized and Non-Urbanized Area)**					
	**No. Cases**	**Population (inhab.)**	**Surface (km^2^)**	**Cumulative Incidence Rate**	**Contagion Rate**	**Density of Infected/km^2^**
3 August 2020 to 16 August 2020	393	574,654	395	68	-	1
17 August 2020 to 30 August 2020	1048			182	2.67	2.7
**Periods**	**Urbanized Area. Territory Outside Focuses of Contagion**					
	**No. Cases**	**Population (inhab.)**	**Surface (km^2^)**	**Cumulative incidence rate**	**Contagion rate**	**Density of Infected/km^2^**
3 August 2020 to 16 August 2020	208	508,458	61.1	41	-	3.4
17 August 2020 to 30 August 2020	251	369,493	55.3	68	1.21	4.5
**Periods**	**Urbanized Area. Territory within Focuses of Contagion**					
	**No. Cases**	**Population (inhab.)**	**Surface (km^2^)**	**Cumulative Incidence Rate**	**Contagion Rate**	**Density of Infected/km^2^**
3 August 2020 to 16 August 2020	185	36,106	1.9	512	-	96.3
17 August 2020 to 30 August 2020	797	152,762	7.8	522	4.31	102.1

**Table 6 ijerph-18-03145-t006:** Quantification of the degree of spatio-temporal permanence of the contagion focuses in two successive bi-weekly periods.

Parameter	No. of Focuses	Surface (km^2^.)	Surface (has.)	Infected
Phase 1 and 2 accumulated focuses	73	9.71	971	982
Focuses phase 1	22	1.92	192	185
Focuses phase 2	51	7.79	779	797
Focuses intersection phase 1 and 2	20	1.35	135.5	170
Focuses intersection phase 1 extended (250 m buffer) with phase 2 focuses	22	4.94	494.34	464
New focuses of phase 2	31	2.15	215.86	246

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
