# Peer review of "The Neighborhood Contagion Focus as a Spatial Unit for Diagnosis and Epidemiological Action against COVID-19 Contagion in Urban Spaces: A Methodological Proposal for Its Detection and Delimitation"

_ijerph, 2021, doi:10.3390/ijerph18063145_

Round 1

Reviewer 1 Report

This paper assessed spatial distribution of COVID-19 cases over 14 days and examined the transition from one period to the other. The methods are appropriately used, and the results are very well presented. I hope my following comments are useful considerations to the authors.

  1. The authors used incidence in 14-day period from health authorities in this paper. I am just curious that are the recovered data appropriately recorded at weekly basis? Did you exclude recovered case from the incidence data?
  2. The authors used 1-3 has as spatial units. I wonder how large is exactly 1 ha? This concept may not be straight forward to some readers. It is hard to interpret ha and compare with size with Administrative units.
  3. The authors presented R index. Is it the same thing as nearest neighbor index?
  4. In section 3.1, I would suggest to use the wording “null hypothesis” instead of “hypothesis 0”.
  5. In table2, I would suggest to use “P<0.01” or “P<0.0001” to replace “P=0” if P value is highly significant.
  6. Due to the fact that many of the cases are detected within the same family, the separation is clearly not normally distributed. Would it be more informative to present Q1, median and Q3 instead of mean separation as mean is not always a good measure. Also, I don't think it is a good idea to delete 0 to generate a new mean. It is kind of a cherry picking from the original data points. I suggest to keep 0 and use median instead of mean under this situation.
  7. In table 6, I feel it is hard to perceive how large is a 1 ha in real life and to compare the results to those aggregated at administrative units. The results will be more informative to the government and the general public if actual sizes in table 6 are presented.

Author Response

REPLY TO REVIEWER 1 SUGGESTIONS.

The authors have incorporated all the modifications indicated by reviewers 1 and 2, and they are grateful for both the correct and interesting suggestions, which improve the quality and expressiveness of the article, and the respectful attitude used in the requirements.

The authors used incidence in 14-day period from health authorities in this paper. I am just curious that are the recovered data appropriately recorded at weekly basis? Did you exclude recovered case from the incidence data?

The recovered data has not been excluded from incidence data becouse, in the considered time period (1 week), the chances of recovery of those afected are very low, and no more than 2 consecutive weekly periods have been considered.

The authors used 1-3 has as spatial units. I wonder how large is exactly 1 ha? This concept may not be straight forward to some readers. It is hard to interpret ha and compare with size with Administrative units.

 One hectare (ha) equals 10,000 m² (100x100 meters)

 The equivalence between hectares and square meters have been clarified in the text.

 At the suggestion of reviewer 1, the dimensions of one hectare are specified in a footnote to the table, and the measurements used are expressed, in addition, to square meters (in a new column of the table).

The authors presented R index. Is it the same thing as nearest neighbor index?

Yes, the R index is the main indicator of nearest neighbors method.

In section 3.1, I would suggest to use the wording “null hypothesis” instead of “hypothesis 0”.

 The wording has been changed in the text (line 535)

In table2, I would suggest to use “P<0.01” or “P<0.0001” to replace “P=0” if P value is highly significant.

 Modification made to the text, table 2 line 336

Due to the fact that many of the cases are detected within the same family, the separation is clearly not normally distributed. Would it be more informative to present Q1, median and Q3 instead of mean separation as mean is not always a good measure. Also, I don't think it is a good idea to delete 0 to generate a new mean. It is kind of a cherry picking from the original data points. I suggest to keep 0 and use median instead of mean under this situation.

At the suggestion of reviewer 1, the parameters Q1, Q2 and Q3 have been calculated for each period considered (table 4, line 397)

An aditional comment has been added on this topic (line 440-443)

In table 6, I feel it is hard to perceive how large is a 1 ha in real life and to compare the results to those aggregated at administrative units. The results will be more informative to the government and the general public if actual sizes in table 6 are presented.

At the suggestion of reviewer 1, the measurements used has been expressed to square kilometers (table 6, line 380)

We would like to clarify that hectare is a unit of very frequent use in the Spanish case, and it is used in a large part of the official statistics. In any case, we have add the equivalence to meters to make the unit of measurement more expressive    

Reviewer 2 Report

I found the article very interesting and well conducted. All section are well organized and aims are well specified. All figures anb tables are usefull and improve the quality of the article.  I suggest to consider also the air quality  of Malaga and if GIS can help also by point of view, this can be another variable in the increase of susceptibility of people  to the contagion. This is only a suggestion but it can be of  interest the readers and it can improve again of the  already good article.
For the rest, I consider this is a very well article.

Author Response

REPLY TO REVIEWER 2 SUGGESTIONS.

The authors have incorporated all the modifications indicated by reviewers 1 and 2, and they are grateful for both the correct and interesting suggestions, which improve the quality and expressiveness of the article, and the respectful attitude used in the requirements.

I found the article very interesting and well conducted. All section are well organized and aims are well specified. All figures anb tables are usefull and improve the quality of the article.  I suggest to consider also the air quality  of Malaga and if GIS can help also by point of view, this can be another variable in the increase of susceptibility of people  to the contagion. This is only a suggestion but it can be of  interest the readers and it can improve again of the  already good article.
For the rest, I consider this is a very well article.

  • Although the relationships of the atmospheric characteristics and air quality of a place with the transmission of Covid-19 is a subject still under study and with uncertain results (Hurtado et al., 2021), Linares-Gil et al. (2020), among others, we are currently advancing in the research along the lines that the reviewer rightly suggests. Following authors such as Zoran et al. (2020), or Coccia (2020), factors such as proximity to the sea, the level of pollution, the slowing of the winds or insolation could be related to the transmission. The problem is that these are factors that vary on a scale of less precision than that used in the present investigation. The levels of the factors indicated are practically homogeneous in the whole of the space of the city, so the statistical correlation is not very expressive.
  • In any case, we are currently trying to associate the construction typologies of buildings and their surroundings with an index of greater or lesser susceptibility to contagion due to the environmental characteristics they generate. This is a complex process that is the subject of a specific publication in the near future.

References:

Coccia, M. (2020). Factors determining the diffusion of COVID-19 and suggested strategy to prevent future accelerated viral infectivity similar to COVID.  Science of the Total Environment 729:138474. https://doi.org/10.1016/j.scitotenv.2020.138474

Hurtado-Díaz, M.; Cruz, C.J.; Blanco-Muñoz, J, et al. (2021) Revisión rápida de los efectos de la variación de la temperatura y la humedad en la morbilidad y mortalidad por Covid-19. Salud Publica de México, 2021; 63 (1): 120-125

Linares-Gil, C. & Diaz-Jimenez, J., & Grupo de Análisis Científico del Coronavirus del ISCIII (GACC-ISCIII) (2020). Clima, temperatura y propagación de la covid-19. Repositorio Institucional de Salud. Gobierno de España. Ministerio de Ciencia e Innovación. http://hdl.handle.net/20.500.12105/9635

Zoran, M.A.; Savastru, R.S.; Savastru, D.M.; Tautan M.N. (2020). Assessing the relationship between ground levels of ozone (O3) and nitrogen dioxide (NO2) with coronavirus (COVID-19) in Milan, Italy. Science of the Total Environment, 740, Oct 20. doi: https://doi.org/10.1016/j.scitotenv.2020.140005